# An experimental target-based platform in yeast for screening *Plasmodium vivax* deoxyhypusine synthase inhibitors

**Suélen Fernandes Silva**[1,2,3], **Angélica Hollunder Klippel**[3,4], **Sunniva Sigurdardóttir**[1], **Sayyed Jalil Mahdizadeh**[1], **Ievgeniia Tiukova**[5], **Catarina Bourgard**[1,6], **Luis Carlos Salazar-Alvarez**[6], **Heloísa Monteiro do Amaral Prado**[3], **Renan Vinicius de Araujo**[3], **Fabio Trindade Maranhão Costa**[6], **Elizabeth Bilsland**[7], **Ross D. King**[5], **Katlin Brauer Massirer**[3], **Leif A. Eriksson**[1], **Mário Henrique Bengtson**[3,8], **Cleslei Fernando Zanelli**[2,4], **Per Sunnerhagen**[1] *

**1** Department of Chemistry and Molecular Biology, University of Gothenburg, Göteborg, Sweden, **2** Chemistry Institute, São Paulo State University - UNESP, Araraquara, São Paulo, Brazil, **3** Center for Medicinal Chemistry - CQMED, Center for Molecular Biology and Genetic Engineering - CBMEG, Universidade Estadual de Campinas, Campinas, São Paulo, Brazil, **4** School of Pharmaceutical Sciences, São Paulo State University—UNESP, Araraquara, São Paulo, Brazil, **5** Department of Life Sciences, Chalmers, Göteborg, Sweden, **6** Laboratory of Tropical Diseases, Institute of Biology, Universidade Estadual de Campinas - UNICAMP, Campinas, São Paulo, Brazil, **7** Department of Structural and Functional Biology, Institute of Biology, Universidade Estadual de Campinas - UNICAMP, Campinas, São Paulo, Brazil, **8** Department of Biochemistry and Tissue Biology, Institute of Biology, Universidade Estadual de Campinas, Campinas, São Paulo, Brazil

* per.sunnerhagen@cmb.gu.se

**Data Availability Statement:** The PvDHS-docked best poses, a table with description of the compounds selected by virtual screening and a

## Abstract

The enzyme deoxyhypusine synthase (DHS) catalyzes the first step in the post-translational modification of the eukaryotic translation factor 5A (eIF5A). This is the only protein known to contain the amino acid hypusine, which results from this modification. Both eIF5A and DHS are essential for cell viability in eukaryotes, and inhibiting DHS is a promising strategy to develop new therapeutic alternatives. DHS proteins from many are sufficiently different from their human orthologs for selective targeting against infectious diseases; however, no DHS inhibitor selective for parasite orthologs has previously been reported.

Here, we established a yeast surrogate genetics platform to identify inhibitors of DHS from *Plasmodium vivax*, one of the major causative agents of malaria. We constructed genetically modified *Saccharomyces cerevisiae* strains expressing DHS genes from *Homo sapiens* (HsDHS) or *P. vivax* (PvDHS) in place of the endogenous DHS gene from *S. cerevisiae*. Compared with a HsDHS complemented strain with a different genetic background that we previously generated, this new strain background was ~60-fold more sensitive to an inhibitor of human DHS.

Initially, a virtual screen using the ChEMBL-NTD database was performed. Candidate ligands were tested in growth assays using the newly generated yeast strains expressing heterologous DHS genes. Among these, two showed promise by preferentially reducing the growth of the PvDHS-expressing strain. Further, in a robotized assay, we screened 400 compounds from the Pathogen Box library using the same *S. cerevisiae* strains, and one compound preferentially reduced the growth of the PvDHS-expressing yeast strain.

table reporting the datasets tested in silico are available as free download at Zenodo.org, https://doi.org/10.5281/zenodo.10006188. DNA sequencing files from cloning constructs and genes integrated into the CAN1 locus, mCherry, Sapphire, and into the DYS1 locus, HsDHS and PvDHS are available at Zenodo.org, https://doi.org/10.5281/zenodo.10053251.

**Funding:** This work was supported by grants from the Swedish Research Council (2016 05627, 2019-03684, and 2021-03667), the Swedish Cancer Fund (22-2014), and the Swedish Foundation for International Cooperation in Research and Higher Education (BR2018-8017) to PS. SFS was the recipient of a fellowship from CAPES PRINT-Brazil (88887.570728/2020-00) and from CNPq (141241/2019-5 and 380882/2023-0). AHK was the recipient of PhD fellowships from CAPES (8887.161266/2017-00), CNPq (380893/2023-1), and from Fundação de Amparo à Pesquisa do Estado de São Paulo (FAPESP) (2018/16672-1 and 2019/24812-0). EB was supported from FAPESP (2015/03553-6) and CAPES (88887.304810/2018-00). RVA was the recipient of FAPESP scholarships (2022/10512-8 and 2023/16654-1). KBM and MHB were recipients of an INCT grant (CNPQ 465651/2014-3 /FAPESP 2014/50897-0). FTMC acknowledges support from FAPESP (2017/18611-7 and 2018/07007-4). LCSA was funded by FAPESP (process 2023/07805-6). The funders had no role in the study design, data collection and analysis, decision to publish, or preparation of the manuscript.

**Competing interests:** The authors have declared that no competing interests exist.

Western blot revealed that these compounds significantly reduced eIF5A hypusination in yeast. The compounds showed antiplasmodial activity in the asexual erythrocyte stage; $EC_{50}$ in high nM to low μM range, and low cytotoxicity.

Our study demonstrates that this yeast-based platform is suitable for identifying and verifying candidate small molecule DHS inhibitors, selective for the parasite over the human ortholog.

## Author summary

The enzyme deoxyhypusine synthase (DHS) modifies the protein eIF5A, essential for translating proteins from mRNA. Without this modification, cells cease protein production and die. DHS is an anticancer drug target. Here, we explore it for targeting the malaria parasite, *Plasmodium vivax*. Using computer-based and robotized searches among large sets of small molecules previously shown to kill *P. vivax* and other pathogens, we found candidate DHS inhibitors. These candidates were tested in genetically engineered yeast strains expressing either human or *P. vivax* DHS. The strains were also modified to uptake external molecules and labeled with a fluorescent protein for quantification. Three of the candidate molecules selectively inhibited growth of the yeast strain expressing *P. vivax* DHS, and blocked eIF5A modification in that strain. These molecules were effective against malaria parasites in human red blood cells grown in the laboratory, including parasite strains resistant to the commonly used antimalarial drug chloroquine, with minimal toxicity to human cells. We demonstrate a synthetic biology method to identify small molecules selective for *P. vivax* DHS, offering potential starting points for antimalarial drug development.

## Introduction

Annually, ~ 250 million cases of malaria are reported, predominantly in low and middle-income countries, resulting in over 600,000 deaths [1]. Despite increased investments in malaria research and disease control over the past century, transmission rates remain high. Furthermore, the prognosis is expected to worsen due to the vector's rapid range shift and its adaptability to urban environments, driven by climate change [2]. Some existing antimalarial drugs suffer from side effects and limited efficacy. Importantly, resistance to available antiplasmodial drugs is rapidly rising worldwide, emphasizing the urgent need for discovery of new and effective targets [3].

Efforts in malaria control have emphasized drug development targeting the responsible parasites, *Plasmodium spp*. The two main agents of malaria are *P. falciparum* and *P. vivax*, where *P. falciparum* accounts for the majority of malaria-related deaths. However, persistent *P. vivax* infections in the liver pose significant challenges to both treatment and disease elimination efforts, leading to high morbidity rates [4]. While continuous culture of *P. falciparum* in its intraerythrocytic stage is possible, there is currently a shortage of continuous *in vitro* culturing system for *P. vivax*; experimentation requires constant replenishment of parasites from patient samples. This prompts the exploration of alternative approaches for assaying potential *P. vivax* drug target proteins. Surrogate genetics systems in the yeast *Saccharomyces cerevisiae* emerges as an attractive option due to the facile genetic engineering in this organism [5–7].

Target-based drug discovery strategies have proven to be effective in developing new effective medicines against infectious diseases [8]. One promising target for investigation is the enzyme deoxyhypusine synthase (DHS), which catalyzes the first step of post-translational modification of the eukaryotic translation factor 5A (eIF5A). In this process, the aminobutyl group is transferred from the polyamine spermidine to a specific lysine residue in eIF5A (K50 in *H. sapiens* eIF5A) [9], forming deoxyhypusine. Subsequently, the enzyme deoxyhypusine hydroxylase (DOOH) hydroxylates deoxyhypusine, leading to the biosynthesis of the unique amino acid hypusine. Notably, eIF5A is the only protein identified to contain this distinctive amino acid [9,10].

Furthermore, both eIF5A and DHS are essential for cell viability in eukaryotes [11–13], including pathogens [14–20]. Hypusinated eIF5A plays a role in various pathologies and physiological processes, contributing to the regulation of cancer [21–24], immune-related diseases [25], diabetes [26,27], neurological disorders [28], aging [29], and the replication of certain viruses [30].

Small molecules capable of modulating mammalian DHS activity through distinct mechanisms, including competitive [31] and allosteric inhibition [32], have already been described. These known inhibitors are often considered as candidates for future anti-cancer therapies [33,34]. Given that DHS is essential, it would be an attractive target for antiparasitic drugs, provided such drugs would be sufficiently selective for the parasite ortholog. However, inhibitors selective for parasite DHS are notably absent. This highlights the urgency to develop novel, targeted inhibitors specific for DHS from eukaryotic pathogens.

Target-based drug discovery campaigns typically rely on *in vitro* biochemical assays, which require protein purification and lack a cellular context [8,35,36]. Alternatively, designing systems with genetically modified *S. cerevisiae* strains can provide efficient platforms for target-based drug discovery within a eukaryotic cell [5–7,37–39].

In this work, we established a yeast target-based platform dedicated to the identification of novel inhibitors of *P. vivax* DHS. We genetically engineered isogenic *S. cerevisiae* strains to differ only in their sole source of DHS, using either the orthologous gene from *P. vivax* or *H. sapiens*. Further enhancements were introduced into these strains to establish a robust system suitable for high-throughput screening, including expressing the target genes from a weak regulatable promoter, and genomic integration of genes encoding fluorescent proteins for precise quantification of cell proliferation. Additionally, genes related to pleiotropic drug resistance were deleted, to facilitate the entry of external molecules into the cells. Combined, these modifications resulted in a strain background ~60-fold more sensitive to the human DHS inhibitor GC7, compared to a human DHS-complemented strain previously generated by us [7] which lacks these genome editions. Using this system, we here identify three promising new small molecules preferentially targeting *P. vivax* DHS over its human counterpart. They are active *in vitro* against erythrocytic *Plasmodium*, and display low cytotoxicity. These newly discovered candidate inhibitors for *P. vivax* DHS can serve as initial leads for the development of potent inhibitors against the *P. vivax* DHS protein. Moreover, the system we have established for *P. vivax* DHS will be applicable to DHS from other parasites and to other parasitic drug targets.

## Results

### Homology modeling of *Plasmodium vivax* DHS

Currently, no 3D protein structure is available for *P. vivax* DHS (PvDHS). To further explore the structural features of *P. vivax* DHS and conduct *in silico* screening of small molecule libraries for potential enzyme inhibitors of this enzyme, we used the YASARA [40] software to generate homology models of PvDHS.

It is worth noting that all previously reported DHS structures are tetramers [7,41,42]. Moreover, DHS in *B. malayi* [7] and *H. sapiens* [32,41] are encoded by a single gene and exist as homotetramers. By contrast, DHS in trypanosomatids are heterotetramers, composed of two DHS paralogs [42]. Considering the structural similarities and the single-gene encoded PvDHS, we generated homotetramer models for this protein. Two models were selected for further investigation in our studies (S1 Fig).

The first model, PvDHS-model 1 (S1A Fig), was constructed based on the deposited structure of *H. sapiens* DHS PDB ID: 6P4V [32], while the second model, PvDHS-model 2 (S1B Fig), was based on *H. sapiens* DHS PDB ID: 6PGR [32]. We developed these two models (S1A and S1B Fig) because 6P4V represents the structure in complex with the cofactor NAD$^+$ and the competitive inhibitor GC7, while 6PGR is in complex with a ligand that induces a conformational change in which an α-helix is unfolded, forming a loop structure. Utilizing these models, our objective was to identify both competitive and/or allosteric inhibitors for PvDHS. SWISS-MODEL reported QMEAN scores for PvDHS-model 1 and PvDHS-model 2 were -2.62 and -3.45, respectively. In addition, the predicted local similarity based on the residue number revealed that residues from both binding sites were located in regions with estimated good quality (score $\geq$ 0.6) (S2 Fig).

The enzyme core of PvDHS shares many features with its human homologue. In fact, the PvDHS protein sequence exhibits 59% similarity with the human DHS and 60% with *S. cerevisiae* DHS (Fig 1). Moreover, the residues in both the GC7 binding site and the allosteric site are conserved between HsDHS, ScDHS and PvDHS (as illustrated in Fig 1). However, PvDHS differs from the human and yeast DHS in their overall length. PvDHS has a total extension of 455 amino acids, making it 86 amino acids longer than HsDHS and 68 longer than ScDHS. Notably, these unique insertions are mainly located from the N-terminal to the middle of the protein extension (Fig 1). The larger insertion (residues 224 to 263) aligns with residues that form an alpha helix in the human enzyme (α8 in Fig 1). Another small insertion, located from residues 100 to 107, aligns with α3 in the human DHS. Currently, it remains unclear whether or how these extensions impact the PvDHS protein.

## Molecular docking

The compound library from the ChEMBL-NTDs database (https://chembl.gitbook.io/chembl-ntd/) was chosen for virtual screening of ligands toward PvDHS. This database comprises several compound libraries that have been tested in previous campaigns for neglected tropical diseases (NTDs) and *Plasmodium* drug discovery campaigns [43,44]. Initially, the dataset was manually curated and filtered, primarily selecting sets that had been previously filtered for compounds tested and found to be active in phenotypic assays against parasites, except for dataset 23 which was not pre-filtered. The datasets chosen for this study are listed in the Zenodo repository. The molecular library prepared from the ChEMBL-NTD database comprised 212,736 structures, accounting for different possible protonation states for each compound.

Given the lack of specific inhibitors for *P. vivax* DHS, we predicted the binding sites based on co-crystallized ligands from human DHS structures (PDB IDs: 6P4V and 6PGR) [32], which served as the basis for the PvDHS models. We defined the binding residues for the known eukaryotic DHS spermidine mimic inhibitor, GC7 (PDB ID: 6P4V) [32], and of the human DHS allosteric ligand, 6-bromo-n-(1h-indol-4-yl)-1-benzothiophene-2-carboxamide (PDB ID: 6PGR) [32] to delineate the anchorage sites for the compounds to be screened. Molecular docking was performed in both the predicted binding sites of PvDHS, and the identified promising ligands were also docked into the HsDHS structures.

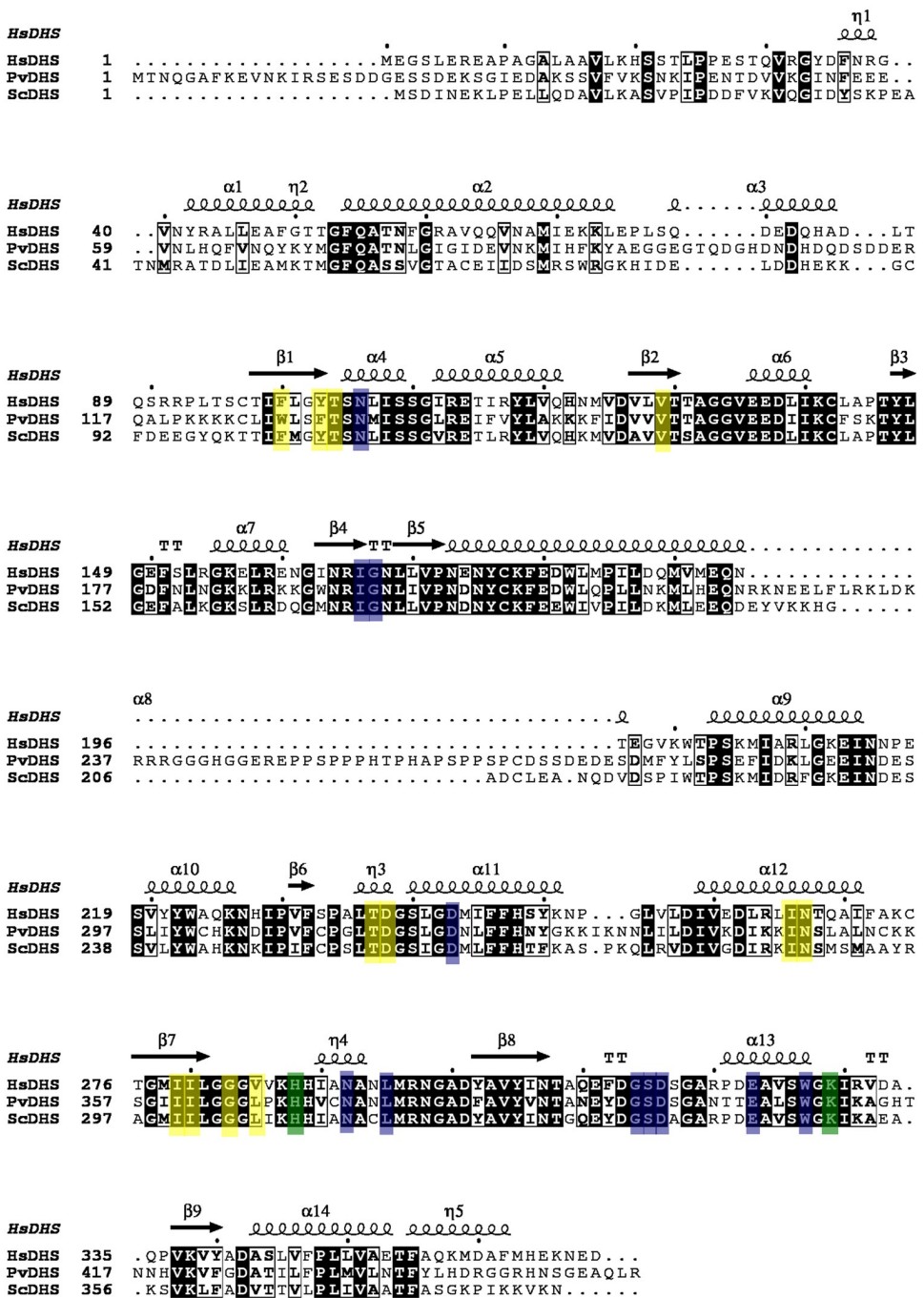

**Fig 1. Structure-based sequence alignment of *H. sapiens* DHS (HsDHS), *P.vivax* DHS (PvDHS) and *S. cerevisiae* DHS (ScDHS).** Residues indicated by a light-colored background participate in the orthosteric binding site (blue), allosteric binding site (yellow), or both (green). Background coloring for PvDHS and ScDHS follows that of the HsDHS proteins. Residues indicated by a black background or framed in a box are conserved in all DHSs analyzed here. The secondary structure (α-helices and β sheets) shown in the top line are for HsDHS. Protein sequences and structures used in the alignment were: HsDHS (UniProt ID P49366, PDB ID 6P4V), PvDHS (UniProt ID Q0KHM1) and ScDHS (UniProt ID P38791).

Before performing the virtual screening, we validated the robustness of the docking protocol by re-docking the co-crystallized ligands, GC7 and 8XY [32] into the human DHS structures (PDB IDs 6P4V and 6PGR, respectively) (S3 Fig). The docking poses of both compounds precisely aligned with the crystal structure, with RMSD values of 0.205 Å for GC7 and 0.208 Å for 8XY, respectively (S3 Fig). The corresponding docking scores were approximately –6.0 kcal/ mol for both compounds. As these ligands are known potent inhibitors of DHS (GC7 $IC_{50}$ = 0.14 μM and 8XY $IC_{50}$ = 0.062μM) [32,45], this value was used as a cutoff threshold to identify new promising ligands, as described in the Methods section.

Following the docking strategy (as described in the Methods section), we identified 100 and 85 promising hits (docking scores < - 6 kcal/mol) from the prepared structures in the ChEMBL-NTD database towards the orthosteric site (GC7 binding site) and allosteric site (8XY binding site) of PvDHS, respectively. The promising hits were also docked into the orthosteric or allosteric binding sites of the *H. sapiens* DHS protein structures, to assess the compound selectivity *in silico*.

Based on the docking score values in PvDHS compared to HsDHS, free energy of binding values, and commercial availability, nine compounds (named N1 to N9), were selected as promising ligands for the PvDHS to be tested in yeast assays. The docking scores and free energy values for the ligands in both predicted binding sites of PvDHS and in HsDHS are listed in Table 1.

According to the docking results, compounds N1 to N6 (N1_o to N6_o in Table 1) were predicted to bind into the orthosteric site. As anticipated, similar to GC7 (S3 Fig), these compounds were positioned at the interface between subunits A and B of PvDHS (S4A–S4F Fig). The interaction profiles between the ligands N1_o to N6_o and the surrounding residues in subunits A and B within the active site of PvDHS are illustrated in S4A–S4F Fig. By contrast, compounds N7 to N9 (N7_a to N9_a in Table 1) were predicted to bind to the allosteric site of PvDHS (Table 1 and S4G–S4I Fig). The structures with the docked compounds are available in the Zenodo database.

## Establishment of a *S. cerevisiae* surrogate genetics platform for discovery of *P. vivax* DHS inhibitors

The yeast *S. cerevisiae* shares conserved pathways with other branches of the eukaryotic phylogenetic tree, making it a valuable model for studying heterologous enzyme function [36,46].

**Table 1. Docking score values obtained by molecular docking of the compounds in the PvDHS and HsDHS proteins.**

| Compound generic name | PvDHS (orthosteric site) | | PvDHS (allosteric site) | | HsDHS* | |
|---|---|---|---|---|---|---|
| | Docking score (kcal/mol) | Free energy of binding (kcal/mol) | Docking score (kcal/mol) | Free energy of binding (kcal/mol) | Docking score (kcal/mol) | Free energy of binding (kcal/mol) |
| N1_o | -7.685 | -48.32 | -4.872 | -33.58 | -5.086 | -38.60 |
| N2_o | -7.576 | -55.60 | -4.362 | -48.02 | -4.023 | -38.97 |
| N3_o | -7.214 | -44.07 | -5.064 | -48.65 | -3.815 | -42.46 |
| N4_o | -6.373 | -31.69 | -3.672 | -21.09 | -3.906 | -46.89 |
| N5_o | -6.097 | -37.68 | -4.811 | -25.10 | -1.790 | -27.24 |
| N6_o | -6.051 | -32.98 | -3.895 | -31.27 | -4.223 | -43.46 |
| N7_a | -5.888 | -33.24 | -9.081 | -34.35 | -4.611 | -39.12 |
| N8_a | -4.372 | -21.75 | -8.200 | -34.28 | -4.472 | -45.15 |
| N9_a | -6.697 | -40.34 | -7.530 | -47.74 | -4.189 | -35.32 |

*: Compounds were docked into the human structures 6P4V (GC7 binding site) or 6PGR (allosteric site) based on which site yielded a higher docking score. N1_o to N6_o denote compounds docked in the orthosteric site and N7_a to N9_a represent compounds docked in the allosteric site.

Yeast surrogate genetics platforms, involving the functional replacement of essential genes in yeast by heterologous genes, have proven effective for infectious diseases drug discovery [5,37,39]. Furthermore, parallel functional complementation with the human and parasite counterparts of DHS offers a way to pinpoint small molecules that selectively modulate the pathogen's protein [37].

To establish a reliable yeast platform to search for *P. vivax* DHS inhibitors, we constructed a mutant *S. cerevisiae* strain with three primary genetic modifications: 1) deletion of a transporter (Snq2) and transcription factors (Pdr1, Pdr3) involved in pleiotropic drug efflux; 2) insertion of fluorescent markers (mCherry, Sapphire); and 3) substitution of the endogenous gene encoding ScDHS (*DYS1*) with orthologous genes expressing PvDHS or HsDHS from a chromosomal locus. Modifications 2 and 3 were performed using CRISPR-Cas9 techniques, as described in the Methods section. Integrating these modifications into the yeast platform strain's genome is crucial, particularly for large-scale experiments. This ensures more consistent expression of the newly integrated gene, thereby enhancing the reliability and reproducibility of our assay strategy [47].

We performed the genetic modifications using a yeast strain with deletions of *PDR1*, *PDR3* and *SNQ2* (HA_SC_1352control [48], S2 Table). *PDR1* and *PDR3* (Pleiotropic Drug Resistance) encode redundant transcription factors positively regulating the expression of various membrane transporters involved in pleiotropic drug resistance, while *SNQ2* (Sensitivity to NitroQuinoline-oxide) encodes a plasma membrane ATP-binding cassette (ABC) transporter [49]. Deleting these genes simultaneously sensitizes yeast to a broad range of compounds, without compromising robustness [50].

For each strain, we inserted genes encoding different fluorescent proteins, mCherry or Sapphire, into the chromosomal *CAN1 locus*, to serve as markers for cell abundance. This enables precise monitoring of growth through fluorescence and turbidity measurements. Additionally, chromosomal expression of the markers reduces noise and overcomes copy number variations, compared to expression from plasmids [51]. Engineering the strains with fluorescent tags enables growth competition assays for different parasite targets with higher precision than using turbidity as the readout [48], allowing simultaneous evaluation of parasite target against the human equivalent within the same well [6].

Subsequently, we replaced the endogenous essential gene encoding ScDHS (*DYS1*) with the orthologous DHS genes from *P. vivax* or *H. sapiens* in the genome. The deletion of ScDHS was confirmed by western blot, as no band was detected with the anti-Dys1 antibody [52] for *S. cerevisiae* DHS (S5A Fig). ScDHS replacement was also confirmed by DNA sequencing. Moreover, PvDHS and HsDHS expressed in yeast were capable of catalyzing hypusination of the endogenous yeast eIF5A (S5A Fig), which demonstrates the functional complementation of the strains.

## Validation of the yeast experimental platform for DHS inhibitor identification

Strains expressing HsDHS or PvDHS (SFS04 and SFS05, respectively; S2 Table) were tested for their ability to detect DHS inhibitors. To validate the platform, we conducted growth curves and western blot assays using GC7, a well-known inhibitor of human DHS [31].

In the newly generated strains, DHS expression was under transcriptional control of the *MET3* promoter, which can be repressed by adding methionine to the culture media [53], enabling tunable control of DHS expression. We determined the methionine concentration at which hypusine levels were similar in the PvDHS and HsDHS complemented strains, and

used this information to define the conditions for assaying the compounds selected by *in silico* screening.

The growth of the *dys1Δ* strain expressing HsDHS was analyzed in 0, 60 and 130 μM methionine, and the PvDHS strain growth was assessed in 0 and 7.5 μM methionine (S5C and S5D Fig). The strain expressing PvDHS did not grow in methionine concentrations above 7.5 μM (S5B Fig). As expected from the greater evolutionary distance between *Plasmodium* and yeast, the level of hypusination in the strain complemented by PvDHS (approximately 40% at 0 μM methionine, S5C Fig) was lower than in the strain with HsDHS (approximately 60% at 0 μM methionine, S5C Fig), indicating that HsDHS complements the *dys1Δ* mutation more efficiently than does PvDHS. However, the hypusine levels in both strains were equal when HsDHS expression was regulated by 130 μM methionine (S5C Fig). Thus, we established growth conditions of the pre-cultures at 130 μM methionine for HsDHS and 0 μM methionine for PvDHS.

It has previously been shown that reducing expression of the target protein in yeast sensitizes growth to inhibition by externally added compounds [5]. Therefore, we performed a growth curve assay to examine the effect of GC7 on the human DHS-complemented strain under various methionine concentrations (ranging from 0 to 130 μM). Notably, we observed a significantly more pronounced growth reduction by GC7 at 130 μM methionine (Fig 2A).

To evaluate if the observed phenotype was indirectly associated with a reduction in the final product of the DHS-DOOH reaction, hypusine, we performed western blot analysis (Fig 2B). In addition, the western blot was used to test hypusine levels after addition of GC7 in the PvDHS-complemented strain, in the absence of methionine. This was defined based on the data presented in S5D Fig, indicating that the hypusine levels are similar between HsDHS and PvDHS under these conditions. As shown in Fig 2, treatment with 50 μM GC7 reduced hypusine levels (Fig 2B and 2C).

## Yeast phenotypic assay using robotized platform

Using the Eve robotized platform programmed for incubation, agitation, and plate reading cycles [6], we tested the PvDHS inhibition of the nine compounds selected through *in silico* screening (Table 1). Growth curve assays were performed (S6, S7 and S8 Figs) and metric parameters extracted from each curve. The area under the curve was selected for comparison of different compound concentrations and the DMSO solvent control (S9 Fig).

In this *in vivo* assay, among the nine tested compounds two, N2 and N7, significantly reduced growth of the PvDHS- complemented strain compared to the strain complemented with HsDHS and the wt strain in the same genetic background (Fig 3).

The nine compounds were tested at concentrations ranging from 25 to 200 μM. When growth assays were performed with the PvDHS-complemented strain, six compounds reduced growth: N1, N2, N3, N5, N7 and N8. However, N1, N3, N5 and N8 reduced the growth of the *HsDHS* and wt strains at least to the same extent (S9 Fig). In contrast, N2 and N7 reduced growth of the PvDHS-complemented strain more than for the other strains (Figs 3 and S6–S9), indicating selectivity for PvDHS.

Notably, N2 exhibited dose-dependent inhibition of the PvDHS-expressing strain, achieving a substantial 95% reduction in growth at 200 μM, 65% at 100 μM, 30% at 50 μM, and 13% at 25 μM of the compound (Fig 3). In contrast, this compound did not significantly affect the growth of HsDHS ($p > 0.05$, represented by the red bars in S9 Fig). Furthermore, N2 at 200 μM only led to a modest 17% reduction in the growth of the wt strain, expressing yeast DHS (illustrated by the green points in Fig 3). The selective growth reduction in the strain expressing PvDHS indicates that DHS is a major target of this compound.

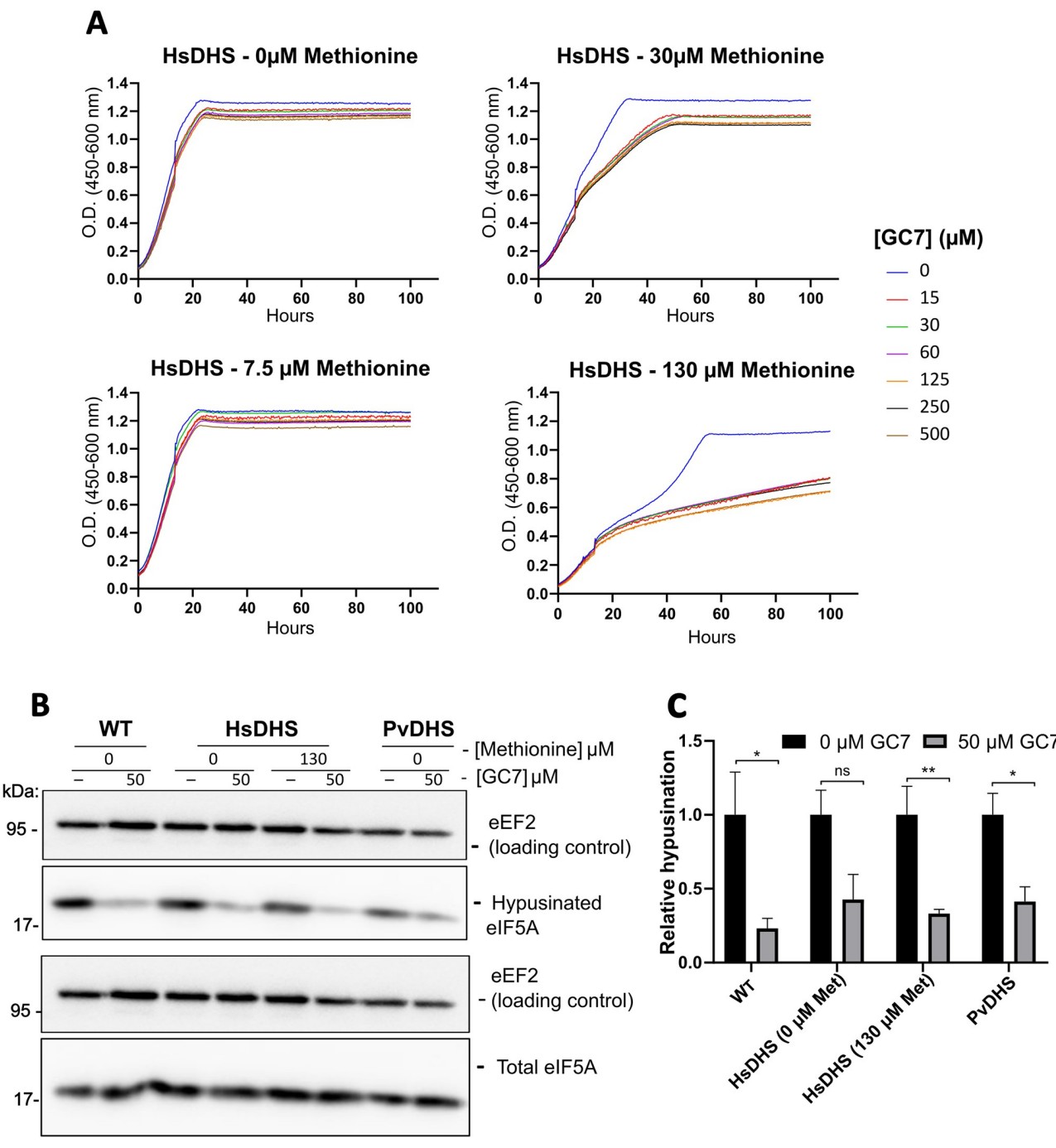

**Fig 2. GC7 inhibits growth of yeast strains, wt or *dys1Δ* complemented by HsDHS or PvDHS.** (A) Comparison of inhibition caused by different concentrations of GC7 (as indicated) in *dys1Δ* yeast complemented by HsDHS (SFS04, S2 Table). The concentration of methionine used to regulate HsDHS expression is indicated in the figure. (B) Western blot detection of hypusination levels after 12 h of treatment with 50 μM GC7 in the wild-type (wt) and PvDHS and HsDHS-complemented strains. The methionine concentrations used in precultures (as mentioned in Material and Methods) are indicated in the figure. (C) Graph reporting the ratio between hypusinated eIF5A and total eIF5A under the conditions indicated in the figure. The bars represent mean of relative hypusination values ((hypusinated eIF5A/ eEF2) / total eIF5A/ eEF2)) ± standard deviation (SD; n = 3). A Student's t-test was performed comparing the compound condition and the solvent control condition where an asterisk (*) indicates that the differences are statistically significant with 95% confidence (p < 0.05); two asterisks (**), 99% confidence (p < 0.01); three asterisks (***), 99.9% confidence (p < 0.001); ns, not statistically significant (p > 0.05).

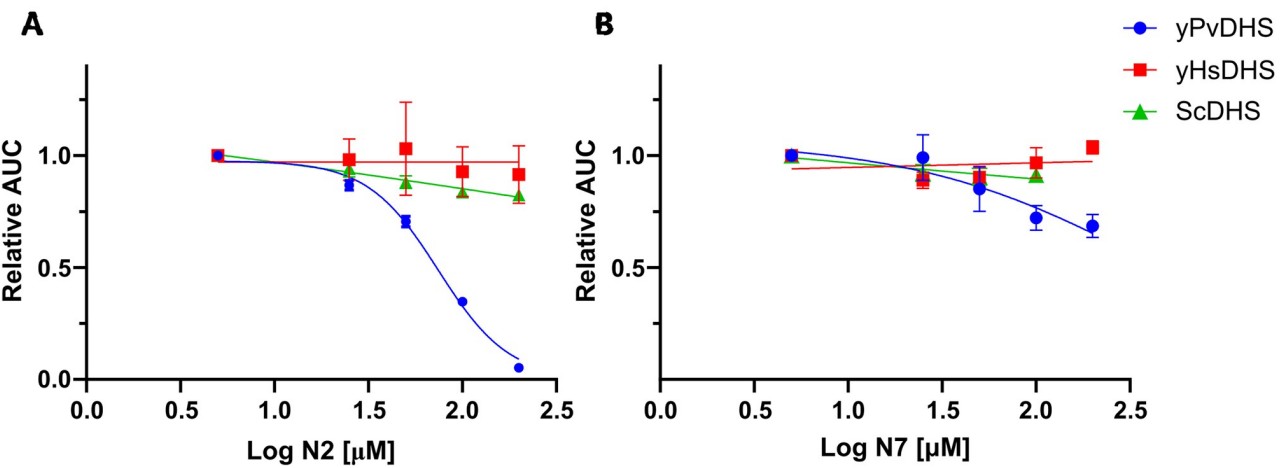

**Fig 3. N2 and N7 detected as potential PvDHS selective inhibitors in the yeast-based assay.** Relative area under the curve (AUC) of *S. cerevisiae dys1Δ* strains complemented by the indicated DHS enzymes, or of isogenic wt (ScDHS). The concentrations of the compounds range from 25 to 200 μM. Statistical significance levels indicated as in Fig 2. Each point represents the mean ± SD of experimental quadruplicates.

Compound N7 yielded a milder growth reduction phenotype, inhibiting approximately 30% of the growth of PvDHS-expressing strain at 100 μM. In comparison, the strain expressing yeast DHS showed 9% inhibition at the same concentration, and no statistically significant inhibition was observed for the HsDHS-expressing strain (Fig 3).

## Screening the Pathogen Box small-molecule library

In an alternative approach, using the same robotized phenotypic screen, we tested 400 compounds from the Pathogen Box small-molecule library (Medicines for Malaria Venture) at a concentration of 25 μM against the PvDHS-complemented strain (SFS05, S2 Table). Most compounds did not affect the growth of the strains. However, 16 compounds (4% of the total molecules in the library) reduced the growth of the PvDHS-complemented strain (S10 Fig). Only few of these 16 compounds were readily commercially available, so we proceeded with confirmatory tests using new batches of compounds PB1 and PB2 (S4 Table). We confirmed growth inhibition of the PvDHS strain by PB1 in a follow-up growth curve assay; showing a dose-dependent response between 50 and 200 μM (Fig 4). With the fresh compound, PB1 reduced 30% of the growth of the PvDHS strain at 200 μM, and by 12% in the HsDHS strain (Fig 4).

## Hypusination is reduced by N2, N7 and PB1 in cells expressing PvDHS, but not HsDHS

Finally, to verify whether the observed growth reduction phenotype in the yeast assay directly correlated with a decrease in the final product (hypusine) of the reaction of which DHS catalyzes the first step, we assessed the hypusination levels in the strains complemented with HsDHS, PvDHS, and the wt strain (SFS04, SFS05, and SFS01, respectively; S2 Table) following a 12 h treatment with the compounds. Specifically, hits from the yeast screening, including N2 (at 100 μM), N7 (at 50 μM), and PB1 (at 100 μM) were tested for their ability to inhibit hypusination in eIF5A (Fig 5).

Compounds N7, N2, and PB1 significantly reduced hypusination levels in the PvDHS-complemented strain by 35% ($p < 0.05$), 60% ($p < 0.05$) and 70% ($p < 0.01$), respectively (Fig 5).

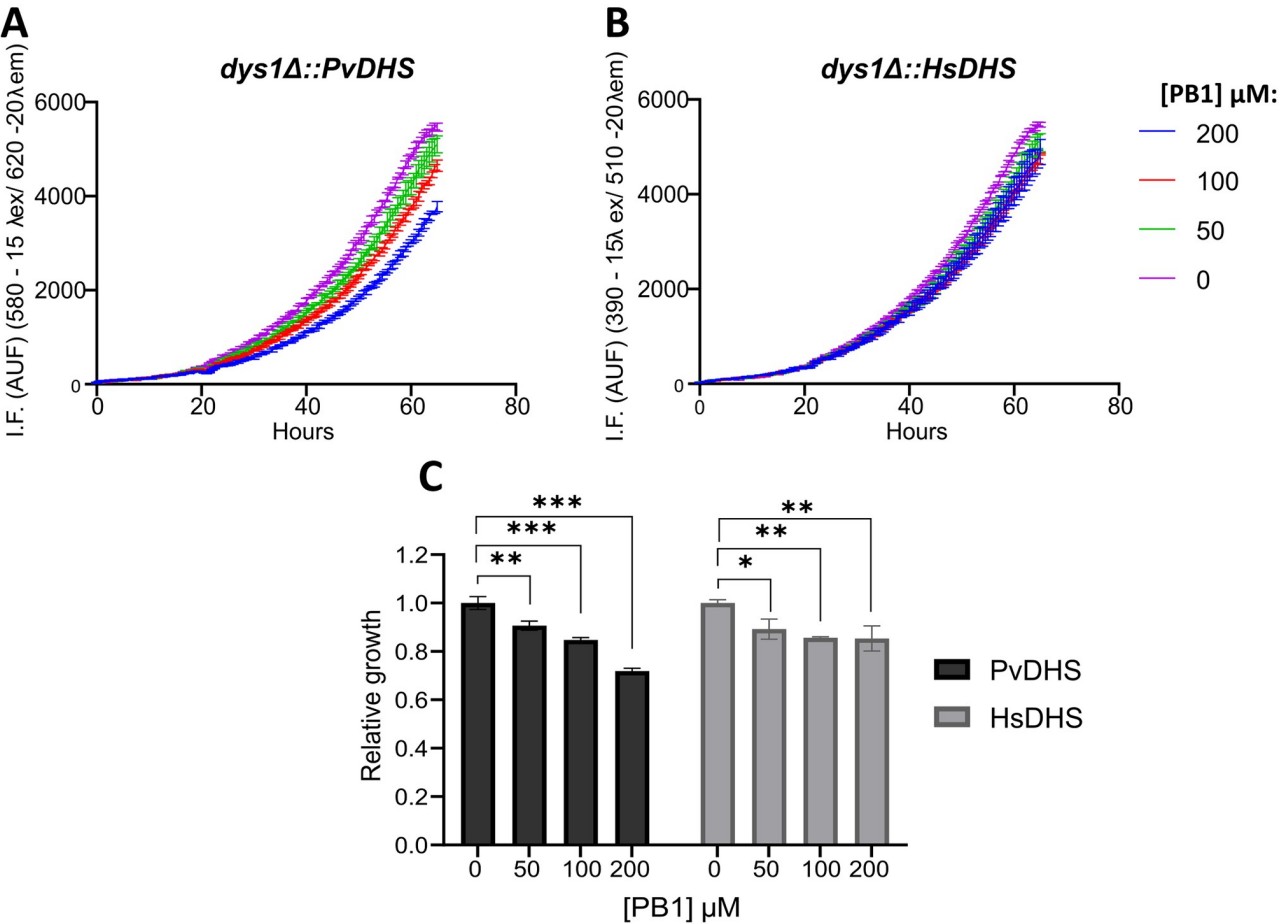

**Fig 4. Relative growth of PvDHS and HsDHS strains in the presence of compound PB1** Growth curves were generated using different concentrations of PB1 (indicated in the figure) in the PvDHS (A) and HsDHS (B) strains. Each point represents the mean ± SD of four experimental replicates. (C) Relative growth score, calculated by the AUC with the compound in relation to the DMSO curve. Statistical significance levels indicated as in Fig 2.

Importantly, no reduction in hypusination was observed in the other tested strains (WT and HsDHS) following a 12 h treatment with these compounds, indicating that the inhibition was specific for PvDHS.

## Antimalarial activity

To evaluate the antimalarial activity of compounds N2, N7 and PB1, we conducted concentration-response assays against the *P. falciparum* chloroquine-sensitive strain *Pf*3D7 and the drug-resistant strain *Pf*Dd2 (S11 Fig). The dose-response assays on the drug-resistant strain *Pf*Dd2 line yielded an $EC_{50}$ of 0.2 ± 0.6 µM (S11A' Fig) for compound N7; 9.4 ± 2 µM (S11B' Fig) for compound N2, and 27 µM for compound PB1 (S11C' Fig and Table 2). The $EC_{50}$ values for strain *Pf*3D7 were 0.3 ± 0.7 µM, 34 ± 2.5 µM and interpolated to more than 20 µM for compounds N7, N2 and PB1, respectively (Table 2).

## Cytotoxicity

To assess the overall toxicity of N2, N7, and PB1, a metabolic viability assay was conducted in human cell cultures using the Presto Blue cell viability reagent (ThermoFisher Scientific),

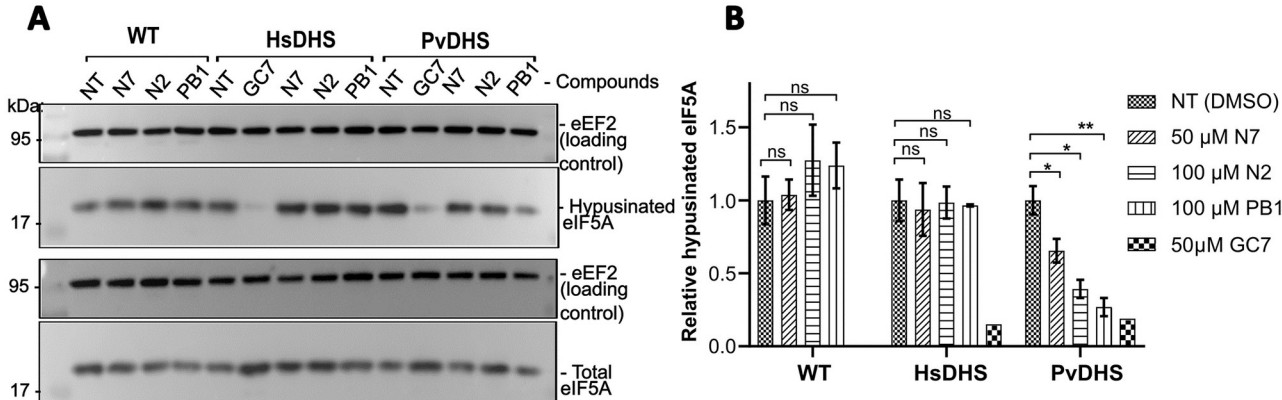

**Fig 5. Hypusination of eIF5A in the presence of PvDHS inhibitor hits.** (A) Detection by western blot of hypusination levels after 12 h of treatment with compounds indicated in the figure, at the following concentrations: GC7 (50 μM), N7 (50 μM) N2 (100 μM) and PB1 (100 μM). The methionine concentrations used in pre-cultures were 130 μM for HsDHS and 0 μM for PvDHS and WT. (B) Graph reporting the ratio between eIF5A hypusinated and total eIF5A under the conditions indicated in the figure. The bars represent the mean relative hypusination values ((hypusinated eIF5A / eEF2) / total eIF5A / eEF2)) ± SD (n = 3); except for GC7 where the bar represents the value for one experiment (n = 1). NT, not treated. A Student's t-test was performed comparing the compound treated condition and the solvent control condition. Statistical significance levels indicated as in Fig 2.

relying on the redox reaction of resazurin. The compounds were tested on two types of human cells: breast cancer cells (MCF7) and hepatocarcinoma cells (HepG2).

Notably, compounds N2 and PB1 demonstrated no significant inhibition of cell viability at the tested concentrations (S12A and S12B Fig) indicating their potential for further drug development. Compound N7 exhibited a moderate toxicity for the tested cell lines, $IC_{50}$ = 32 μM (S12C Fig).

## Discussion

We have here established an *S. cerevisiae* surrogate genetics system for high-throughput screening to identify new parasite-selective inhibitors of *P. vivax* DHS. This approach was a further development of previous yeast surrogate systems designed for target-selective inhibitor searches designed to circumvent challenges associated with *in vitro* growth of parasites [5,38,39]. Utilizing CRISPR-Cas9 homology-directed repair, we have incorporated new features including fluorescent markers and genetic integration of the orthologous DHS genes, greatly enhancing the platform's robustness and providing control over a wider range of target protein expression levels, using the weak regulatable *MET3* promoter.

Our newly generated strain, with DHS expression regulated by the *MET3* promoter and reduced ABC exporter activity, exhibited enhanced sensitivity to external small molecules.

**Table 2. *In vitro* analysis of the antiplasmodial capacity of compounds against *P. falciparum* Dd2 (chloroquine-resistant) and 3D7 (chloroquine-sensitive).**

| ID | $EC_{50}$ (μM) [a] | |
|---|---|---|
| | *Pf*Dd2 | *Pf*3D7 |
| N7 | 0.2 ± 0.6 | 0.3 ± 0.7 |
| N2 | 9.4 ± 2.0 | 34 ± 2.5 |
| PB1 | 27 ± 4.5 | >20 |
| Artesunate | 0.0015 ± 0.0005 | 0.0023 ± 0.001 |
| Chloroquine | 0.075 ± 0.004 | 0.013 ± 0.003 |

[a] $EC_{50}$: half of the maximum inhibitory concentration in *Pf*3D7 and *Pf*Dd2 strains and their respective standard deviations.

This allowed growth inhibition at lower concentrations (15 μM) compared to previous studies using *Leishmania major* and *H. sapiens* DHS- complemented strains lacking these modifications, where 1 mM GC7 was required due to poor compound permeability [7]. This corresponds to a 60-fold improvement of sensitivity. There are limitations to our system, in that the readout proxy (cell proliferation) is only indirectly linked to the primary event (eIF5A hypusination). The dose-response relationship is therefore non-linear, and so the method is semi-quantitative. We can only use it in limited ranges of cell growth phase, inhibitor concentration, eIF5A hypusination etc. However, the method can be used to rank candidate inhibitors for their selectivity for the target protein parasite ortholog.

To ensure the experimental conditions were as comparable as possible for both the HsDHS and PvDHS-complemented strains before testing compounds selectivity, we conducted preliminary assays to establish methionine concentrations yielding similar hypusination levels in each strain. HsDHS and PvDHS showed approximately 60% and 40% hypusination, respectively, at 0 μM methionine. We adjusted the methionine concentrations accordingly, 130 μM for HsDHS and 0 μM for PvDHS, to equalize the levels of hypusinated eIF5A, thus enabling a fair comparison during inhibitor screening.

Here, with this established yeast-based platform, we screened a total of 409 compounds against PvDHS, including 400 molecules from the Pathogen Box library (Medicines for Malaria Venture) and nine compounds selected through virtual screening from a ChEMBL-NTD database. From this screening three compounds targeting PvDHS, based on their differential growth–N2, N7, and PB1 –were identified.

The distinctive amino acid hypusine is the final product of the reaction catalyzed by the enzymes DHS and DOHH in the post-translation modification of eIF5A [13]. Hence, we performed a secondary assay to investigate whether these three compounds—N2, N7 and PB1—decreased the levels of hypusinated eIF5A. All three compounds significantly reduced the levels of *S. cerevisiae* hypusinated eIF5A levels selectively in the strain complemented by PvDHS, indicating preferential inhibition of the *Plasmodium* enzyme.

Our molecular docking studies show that compound N2 binds the predicted orthosteric site of *P. vivax*, while N7 binds to the allosteric site. With a similar ligand interaction profile as GC7, N2 forms hydrogen bonds with the amino acids Gly395A, Glu404A and Lys410A, as well as a hydrophobic interaction with Trp408A. Additionally, unlike GC7, N2 exhibits a halogen bond with Asn192B via the chlorine atom. Predicted interactions for N7 include a halogen bond between the fluorine atom and Phe310A, a hydrogen bond with the side chain of Lys410A and multiple hydrophobic interactions, such as with Trp128A and Ile359A. No protein targets had been previously reported for these compounds.

PB1 was identified through screening the Pathogen box rather than molecular docking. This compound was previously reported to have dual inhibitory activity *in vitro* against the *Mycobacterium tuberculosis* proteins EthR and InhA [54]. We now propose *P. vivax* DHS as a novel target for this small molecule.

We tested these compounds *in vitro* against *P. falciparum* in an erythrocyte-based infection assay. N2 and PB1 exhibited $EC_{50}$ in the low μM range against the drug-resistant *P. falciparum* Dd2 strain, whereas no cytotoxicity could be detected. N7 exhibited $EC_{50}$ in the high nM range against both *P. falciparum* strains tested, and $EC_{50}$ around 90 times higher in the human tested cell lines, demonstrating its potential as an antimalarial agent. The compounds identified showed selective inhibition for *Plasmodium* DHS in comparison to the human enzyme and were able to reduce intracellular levels of hypusine, indicating that DHS is one of the cellular targets of those compounds.

In conclusion, here we report the development of an alternative approach for drug discovery based on *S. cerevisiae* strains genetically engineered for increased sensitivity to external

small molecules and more stable target gene expression, designed against the promising protein target, PvDHS. These principles can be extended to other parasitic organisms and drug targets.

## Materials and methods

### Plasmids and strains

Maintenance and cultivation of strains, preparation of culture media and solutions were conducted following standard protocols [55,56]. Plasmids and *S. cerevisiae* strains used in this study are listed in S1 and S2 Tables, respectively. The sequence of primers used are given in S3 Table.

### Synthetic compounds

The well-established DHS inhibitor, GC7 [31,33,57], and nine compounds (here named N1 to N9), selected from *in silico* screening in this work, were purchased from different suppliers (S4 Table). Stocks of all compounds were prepared in DMSO at a concentration of 50 mM and stored at -20˚C. The extended compound description is presented in S4 Table. Additionally, 400 small molecules from the Pathogen Box library (Medicines for Malaria Venture) were tested. The stocks of these compounds were at a concentration of 10 mM in DMSO (Sigma) and stored at -80˚ C. New batches of two compounds (named PB1 and PB2, S4 Table) from this collection were acquired for confirmatory testing.

### Virtual screening of the ChEMBL-NTD repository

The 3D structures of the DHS protein from *P. vivax* were generated through homology-modelling using the YASARA software [40] and protein sequence retrieved from UniProt database (full length sequence; UniProt ID Q0KHM). Protein models quality were estimated using the SWISS-MODEL Structure assessment online platform [58]. For *H. sapiens* DHS, high-resolution crystallographic structures were readily available (PDB IDs: 6P4V and 6PGR [32]), which were utilized in the screening campaigns. The protein structures underwent preparation with the Protein Preparation Wizard (Schrödinger Release 2021–2: Schrödinger Suite 2021–2 Protein Preparation Wizard) [59] within the Maestro software (Schrödinger Release 2021–2: Maestro, NY, 2021), applying default parameters. Protein preparation included removing water molecules and ligands, addition of hydrogens, assigning bonds and bond orders, completing missing loops and/or side chains using Prime [60,61], and optimizing hydrogen bonding network by adjusting the protonation states of Asp, Glu and tautomeric states of His to match a pH of 7.0 ± 2.0. Subsequently, geometric refinements were conducted using the OPLS4 force field [62] in restrained structural minimization.

Ligand structures were acquired from the ChEMBL-NTD database (https://chembl.gitbook.io/chembl-ntd/) in SDF format, and prepared using the LigPrep tool (Schrödinger Release 2021–2: Schrödinger Suite 2021–2, LigPrep) [59]. Ionization and protonation states were assigned in the pH range of 7.0 ± 2.0 using Epik [63] and energy minimization was performed with the OPLS4 force field [62].

Coordinates corresponding to the amino acids involved in the binding to the DHS inhibitor GC7 [13] and the allosteric inhibitor [32] were defined to generate the docking grid-boxes in the prepared protein structures. Molecular docking was executed using the Virtual Screening Workflow (Schrödinger Release 2021–2: Schrödinger Suite 2021–2, Virtual Screening Workflow Glide) [64], in the following selection stages: top 10% poses selected by Glide HTVS (high-throughput virtual screening) were used in the Glide SP (standard precision); top 10%

poses from Glide SP were used in Glide XP (extra precision); and post-processing employing Molecular Mechanics with Generalized Born and Surface Area (MM-GBSA) method for the free energy of binding calculation using Prime [65].

In this strategy, we initially re-docked the co-crystalized ligands into the corresponding receptor protein upon which the PvDHS models were based. The docking score values of the co-crystalized ligands were employed as a cutoff threshold to consider new compounds as promising hits. Finally, we analyzed the free energy of binding values and the spatial disposition of the ligands within the binding sites to predict whether the compound would fit well within the receptor site.

## Construction of an *S. cerevisiae* DHS surrogate genetics platform

We developed a yeast surrogate genetics platform by adapting previously described CRISPR/Cas9 methodologies [51,66]. *S. cerevisiae* strains with two primary modifications were created: chromosomal integration of sequences expressing the fluorescent proteins mCherry and Sapphire, and replacement of the essential gene *DYS1* (which encodes DHS in *S. cerevisiae*) with DHS coding sequences from *P. vivax* or *H. sapiens*. These were integrated into the *DYS1* locus in the genome of the *S. cerevisiae* working strain.

The genetic background for generating these strains was the *S. cerevisiae* HA_SC_1352control strain [48] (S2 Table). This strain has deletions of *PDR1* and *PDR3*, encoding transcription factors that regulate pleiotropic drug responses, as well as *SNQ2*, a membrane transporter of the ABC family [50].

**Genomic integration of mCherry and Sapphire genes.** The coding sequences for the fluorescent proteins Sapphire and mCherry, under control of the *TDH3* constitutive promoter, were inserted in the *S. cerevisiae* HA_SC_1352control strain into the *CAN1* locus, using CRISPR/Cas9-mediated homology-directed repair [66].

**sgRNA design, assembly and preparation.** The design, assembly, and preparation of the single guide RNA (sgRNA) targeting *CAN1* locus were adapted from the Benchling platform (https://benchling.com/pub/ellis-crisprtools). We selected a previously reported sgRNA targeting *CAN1* [66] (S3 Table). The oligonucleotides containing the guide sequences were annealed and cloned into the *Bsm B*I site of the sgRNA entry vector pWS082 (S1 Table) using Golden Gate assembly protocol [67]. This process replaced the GFP coding sequence from pWS082, generating the pWS082-CAN1 sgRNA construct (S1 Table). Half of the reaction mixture was used for transformation into *E. coli* 10-beta (New England Biolabs). White colonies (indicating the absence of GFP) were selected, and the pWS082-CAN1 sgRNA plasmid (S1 Table) was extracted using the GeneJet Plasmid miniprep kit (ThermoFisher Scientific). Accuracy of the cloning was confirmed through DNA sequencing.

**Yeast transformation.** We used a standard yeast transformation protocol [68] incorporating three DNA components: 300 ng of pWS082-CAN1 sgRNA vector, linearized by *Eco R*V digestion; 100 ng of the pWS172 previously digested with *Bsm B*I and 3 μg of PCR-amplified donor DNA. The donor DNA consisted of the mCherry/Sapphire sequences, with the addition of *CAN1* homologous arms. This process generated the yeast strains SFS01 and SFS02 (S2 Table).

**Deletion of *S. cerevisiae DYS1* and replacement by DHS from *H. sapiens* and *P. vivax*.** The coding sequences of DHS from *H. sapiens* (Gene ID: 1725, isoform A, here referred to as HsDHS) and *P. vivax* (Gene ID: 5476136, here referred to as PvDHS) were synthesized with codon usage optimization for expression in *S. cerevisiae* (S1 Text) and amplified by PCR using the oligonucleotides described in S3 Table. Subsequently, they were cloned into the *Bam H*I-*Pst* I sites of the yeast expression vector pCM188-MET3 (S1 Table), resulting in the generation

of the following constructs: pCM188-MET3-HsDHS and pCM188-MET3-PvDHS (S1 Table). For details on all constructs generated in this study, including primer sequences, consult S1 and S3 Tables. The DNA constructs were used in this work for genomic integration protocols and for the yeast-based functional complementation system, and were verified by sequencing.

**sgRNA design, assembly and preparation.** The sgRNA targeting *DYS1* was designed using the online platform ATUM (https://www.atum.bio/eCommerce/cas9) (S3 Table). The assembly and preparation process followed the same protocol described for *CAN1* in this work, resulting in construction of the plasmid pWS082-DYS1 sgRNA.

**Yeast transformation.** The transformation reaction followed the same procedure as the integration of the fluorescent proteins into the *CAN1* locus. However, the DNA components included in the reaction were as follows: 300 ng of pWS082-DYS1 sgRNA vector, linearized by *EcoR*V digestion; 100 ng of the pWS158 previously digested with *BsmB*I; and at least 5 μg of PCR-amplified donor DNA. The donor DNA comprised the MET3pr-DHS sequences from *H. sapiens* or *P. vivax CYCt* cassette, with the addition of *DYS1* homologous arms. These linear fragments were transformed into strains SFS01 and SFS02 (S2 Table), generating strains SFS05 and SFS04 (S2 Table), respectively. To confirm genome integrations, colony PCR was conducted and DNA sequencing.

**Culture conditions.** The *S. cerevisiae* strains used in this work are described in Table 1. Synthetic complete (SC) medium (0.19% yeast nitrogen base without amino acids and ammonium sulfate, 0.5% ammonium sulfate, 2% glucose) with dropout of the appropriate amino acids was used for culture maintenance and assays performance.

## Validation of the platform in the presence of the drug GC7

Growth curves of yeast strains expressing DHS from *H. sapiens* (strain SFS04, S2 Table) or *P. vivax* (strain SFS05, S2 Table) were analyzed in the presence and absence of inhibitors. Prior to testing the inhibition by new compounds, we conducted an initial assay standardization using the well-known inhibitor of the human DHS enzyme, GC7 [31]. The assay was done in the presence of GC7 and different methionine concentrations added to the SC medium. With that, we aimed to evaluate GC7 inhibition across various levels of DHS expression.

To prepare for the experiment, a 5 mL pre-culture was grown at 30˚C under agitation until it reached approximately $1 \times 10^7$ cells/mL. Once this density was achieved, the pre-cultures were diluted to a concentration of approximately $1 \times 10^6$ cells/mL ($OD_{600nm} = 0.2$). The cultures were then incubated under agitation for approximately 3 h until they entered the exponential growth phase ($OD_{600nm}$ ~0.6). Just before the experiment, the cultures were diluted to $OD_{600nm} = 0.1$ in the appropriate SC medium and dispensed in 200 μL aliquots per well in 100-well plates (Honeycomb plates, Labsystems Oy) using the automated OT2 robot pipettor (Opentrons). The strains were then cultivated for 72 h at low agitation, with absorbance values ($OD_{450-600nm}$) measured every 20 min using the Bioscreen C plate reader equipment (Oy Growth Curves Ab Ltd).

## Phenotypic assays using a robotized platform

The strains were cultivated in SC liquid medium, utilizing the optimal methionine concentrations determined through the growth validation assay.

Cultures with a final $OD_{600nm}$ of 0.1 were dispensed into black 384-well plates (Greiner) using the Multidrop Combi reagent dispenser system (ThermoFisher). The compounds selected for virtual screening (S4 Table) were transferred to the plates using the Bravo Liquid Handling platform (Agilent) to final concentrations ranging from 25 to 200 μM for each compound (resulting in a final DMSO concentration of 1.25%) in a total volume of 80 μL.

The cultures were incubated at 30˚C, and their growth was monitored as previously described [6]. An automated platform was programmed for cycles of incubation, agitation, and plate reading using the Eve robot [6]. Fluorescence values (mCherry = 580 nm $\lambda_{excitation}$ / 612 nm $\lambda_{emission}$; Sapphire = 400 nm $\lambda_{excitation}$ / 511 nm $\lambda_{emission}$) and absorbance ($OD_{600nm}$) were measured every 20 min over a period of 3 days using the BMG Polarstar plate reader. The experiments were conducted in two different experiments, each in quadruplicate.

To visually assess growth reduction, growth curves were plotted using the mean of fluorescence values or absorbance, along with their respective standard deviations, derived from quadruplicate measurements. A solvent control sample was included in the same graph as a reference.

A model was fitted, considering logistic growth model equations. The parameters relative to the curves were extracted using the Growthcurver package [69] in the R studio software (Version 2022 12.0 for Windows, Posit Software). The area under the curve (AUC) values were extracted for each curve and selected to calculate a relative score (relative growth = AUC compound / AUC DMSO). Student's t-tests were conducted for four-sample comparisons, comparing the growth in the presence of compound with growth in solvent-only control.

In a different approach, compounds from the Pathogen Box library (Medicines for Malaria Venture) were also tested in the strain complemented by PvDHS with the same methodology. The experiment was carried out in quadruplicate, in one experiment. The curves representing the means were plotted in the R software. Two candidates from this screen were selected for confirmation (S4 Table).

## Evaluation of hypusination levels by western blot assay

The hypusination levels of the complemented strains were evaluated by western blot. The strains were cultured for approximately 14 h in SC medium without methionine. Pre-cultures were then diluted to $OD_{600nm}$ = 0.15 and grown during 10 h in fresh SC medium with different methionine concentrations. Cells were harvested by centrifugation at 4˚C and stored at -80˚C.

For the assays in the presence of inhibitors, after 10 h of cell growth in SC medium with the specified methionine concentration, cells were washed and diluted again to $OD_{600nm}$ = 0.15 in SC medium lacking methionine. Inhibitor candidates were added at a concentration of 100 μM, except for compounds GC7 (inhibition control) and N7, which were added at a concentration of 50 μM. A control for 0% inhibition was also prepared, containing only cells and DMSO. After 12 h of treatment, cells were collected by centrifugation at 4˚C and stored at -80˚C.

Cell pellets were resuspended in 40 μL of lysis buffer (200 mM Tris/HCl, pH 7.0, 2 mM DTT, 2 mM EDTA; 2 mM PMSF) and lysed by agitation for 10 min with glass beads. The cells were clarified by centrifugation at 20,000 × g for 15 min at 4˚C, and the supernatant was collected. Protein quantification was performed using the BCA protein quantification assay (QPro BCA kit, Cyanagen) against a bovine serum albumin (BSA) calibration curve (Sigma Aldrich).

A total of 12 μg of total protein was loaded onto a 12% SDS-PAGE gel and transferred to a methanol pre-activated PVDF membrane using a semi-dry transfer system (Bio Rad, catalog number 1704150) following the equipment's standard protocol. The western blot employed the following antibodies: rabbit polyclonal anti-eIF5A (yeast) antibody at a 1:10,000 dilution; rabbit polyclonal anti-hypusine (Millipore) antibody at a 1:2,000 dilution; and a rabbit polyclonal anti-eEF2 (yeast) antibody at a 1:25,000 dilution (sample loading control) (Thermo-Fisher), followed by a secondary anti-rabbit antibody at a 1:20,000 dilution (Sigma Aldrich).

The proteins of interest were detected using an enhanced chemiluminescence detection system (ECL, Amersham) on the Uvitec photodocumentation system (Alliance). The bands detected in the Western blot were quantified using GelAnalyzer 19.1 software [70]. The bands detected for anti-eIF5A and anti-hypusine were normalized in relation to the loading control bands (anti-eEF2), and the hypusination rate was calculated by dividing the values obtained for hypusinated eIF5A by the values quantified for total eIF5A. An F-test was performed to assess whether the variances were equivalent or different. Based on the known variance, a Student's t-test was conducted to verify statistical significance of the data.

### *P. falciparum* cultures *and in vitro* antiplasmodial activity

The antimalarial activity assay was assessed on *P. falciparum* 3D7 (chloroquine-sensitive), and Dd2 (chloroquine-resistant) [71]. The parasites were cultivated in RPMI 1640 medium (SIGMA-ALDRICH), supplemented with hypoxanthine 50 mg/L, glucose 2 g/L, sodium bicarbonate 2 g/L, $O^+$ red blood cells (RBCs) and $A^+$ human plasma 10%. The cultures were incubated at 37˚C in a low oxygen environment (3% $O_2$, 5% $CO_2$, and 92% $N_2$) as previously described by Trager *et al.* [72]. Drug inhibition assays were performed as previously described [73]. Briefly, synchronizations with 5% D-Sorbitol solution were performed at 48 h intervals before the experiments, to allow incubations with > 90% of the parasites in the ring stage. Assays were performed in a 96-well plate with 1% parasitemia and 2% hematocrit. Tests were carried out at concentration ranges from 0.01 to 80 μM and the $EC_{50}$ values were determined. Artesunate and chloroquine were used as reference antimalarial drugs. After 72 h of incubation, parasitemia was assessed by fluorometry using SybrGreen fluorescent dye [74]. The plates were read in a Varioskan LUX Microplate Reader (ThermoScientific) by fluorescence at 490 nm excitation and 540 nm emission wavelengths. The growth inhibition values were expressed as percentages relative to the drug-free control and $EC_{50}$ value were calculated by plotting Log dosing vs growth inhibition (expressed as percentage of the drug-free control) using GraphPad Prism (version 8.0). The experiments were carried out in three independent assays.

### Evaluation of cytotoxicity in mammalian cell lines

**Cell culture conditions.** The tumorigenic cell lines MCF7 (breast adenocarcinoma, ATTC: HTB-22) and HepG2 (hepatocellular carcinoma, ATCC: HB-8065) were cultivated in DMEM (Gibco DMEM, ThermoFisher Scientific) culture media supplemented with 10% fetal bovine serum (Gibco FBS, ThermoFisher Scientific) and 1% penicillin-streptomycin (Pen-Strep, ThermoFisher Scientific). The cell lines were incubated at 37˚C with 5% $CO_2$.

**Evaluation of cytotoxicity using resazurin assay.** Cell viability in the presence of compounds was assessed using the resazurin-based cell viability reagent, Presto Blue (Thermo Fisher Scientific). MCF7 and HepG2 cells were seeded into 96-well plates (TPP) at a concentration of $10^4$ cells/well in 200 μL of culture medium and incubated for approximately 24 h until the cells adhered to the plate. The compounds selected as DHS inhibitor candidates in the yeast assay were added to the plates at concentrations ranging from 0.1 to 100 μM (with a maximum DMSO concentration of 0.4%) and incubated for 24 h. After this, 10% Presto Blue was added, and the cells were further incubated for 1 h. Resorufin fluorescence was measured at an excitation wavelength of 544 ± 15 nm and an emission wavelength of 590 ± 15 nm using the spectrofluorometer POLARstar Omega (BMG Labtech). Each experiment was conducted in duplicate, and the experiments were repeated twice.

Cell viability percentages were determined by the quotient between the fluorescence intensity values of cells treated with the compounds by the values from solvent control cells, multiplied by 100. The concentrations that inhibited 50% of cell growth ($IC_{50}$) were calculated

through non-linear regression. The percentage of cell viability was plotted against the logarithmic concentration of the compounds using GraphPad PRISM software (version 8.0 for the Microsoft Windows operating system), and the results are represented as the mean ± standard deviation.

## Supporting information

**S1 Fig. Homology models of *P. vivax* DHS homotetramer.** (A) Cartoon representation of the PvDHS—model 1 (red) superposed on the HsDHS structure 6P4V [32] (blue). (B) Overlay of PvDHS—model 2 (yellow) superposed on the HsDHS structure 6PGR [32] (magenta). (C) Overlay of α-helix unfolded in PvDHS-model 2 (gray) with the corresponding position in PvDHS-model1 (green).
(DOCX)

**S2 Fig. Local quality assessment of residues in the PvDHS-model 1 and PvDHS-model 2.** The black dashed line indicates the threshold distinguishing poor- and high-quality local similarity regions. Arrows indicate the position of the residues within the binding sites.
(DOCX)

**S3 Fig. Alignment of the re-docked ligands with its original co-crystallized conformation.** Zoomed-out view of GC7 (pink) and 8XY (red) disposition in the tetramer structures 6P4V (A) and PDB ID: 6PGR (C), respectively. B and D show the overlay of the original ligands, GC7 (pink) and 8XY (red) with the ligands after redocking (gray). Different colors represent individual chains in the structure.
(DOCX)

**S4 Fig. Predicted binding mode of the selected compounds in PvDHS.** The binding mode and interaction profile of the orthosteric ligands N1 to N6 (A to F, respectively; in orange, labeled in the figure) and allosteric ligands (N7 to N9, G to H, respectively) in PvDHS residues (gray). The residues from PvDHS that interact with the compounds are identified by the three-letter amino acid code followed by number and chain. Different type of interactions is represented by different colors and lines, and their corresponding meaning are described at the bottom of the figure.
(DOCX)

**S5 Fig. Hypusination of yeast eIF5A by *H. sapiens* and *P. vivax* DHS-complemented strains.** (A) Western blot detection of hypusinated eIF5A and total eIF5A from the *S. cerevisiae* strains wt, *dys1Δ*::*HsDHS* and *dys1Δ*::*PvDHS* (SFS01, SFS04 and SFS05, S2 Table). An antibody specific for *S. cerevisiae* DHS (ScDHS) was used to show the deletion of the endogenous gene in the strains replaced by HsDHS and PvDHS. (B) Inhibition of growth of *P. vivax* DHS-complemented by different methionine concentrations. (C) Western blot detection of hypusination levels after 10 h of methionine treatment (concentrations indicated in the figure). (D) Semi-quantitative graph reporting the percentage of hypusinated eIF5A (hypusinated eIF5A/total eIF5A *100) under the conditions indicated in the figure. Bars represent mean ± standard deviation (n = 3) of the percentage of hypusinated eIF5A.
(DOCX)

**S6 Fig. Growth of yeast DHS deleted strain complemented by PvDHS in the presence of compounds N1 to N9.** The strain used was SFS05 (S2 Table). The growth measurements were carried out in the Eve robot (see Materials and methods) and it is given in arbitrary fluorescence units (AFU) (mean ± SD, n = 4). Cell cultures were grown in SC–met and 1.25% DMSO

or 0, 25, 50, 100 and 200 μM concentrations of the respective compound tested (see legend).
(DOCX)

**S7 Fig. Growth of yeast DHS deleted strain complemented by HsDHS in the presence of compounds N1 to N9.** The strain used was SFS04 (S2 Table). The growth measurements, conducted in the Eve robot (see Materials and methods) are presented in arbitrary fluorescence units (AFU) and shown as mean ± SD, n = 4 replicates). Cell cultures were grown in SC with solvent alone (1.25% DMSO) or varying concentrations (0, 25, 50, 100 and 200 μM) of the respective tested compound, as indicated in the legend.
(DOCX)

**S8 Fig. Growth of yeast wt isogenic strain in the presence of compounds N1 to N9.** The strain used was SFS01 (S2 Table). The growth measurements were carried out in the Eve robot (see Materials and methods) and it is given in arbitrary fluorescence units (AFU) (mean ± SD, n = 4). Cell cultures were grown in SC and solvent alone (1.25% DMSO) or varying concentrations or (25, 50, 100 and 200 μM) of the respective compound tested as indicated in the legend).
(DOCX)

**S9 Fig. Relative area under the curve extracted from growth curves.** Bars represent the relative growth score of S. cerevisiae wt or dys1Δ strains complemented by the PvDHS or HsDHS enzymes. The compound concentrations range from 25 to 200 μM. Statistical significance levels indicated as in Fig 2.
(DOCX)

**S10 Fig. Growth reduction of PvDHS-complemented strain caused by compounds from the Pathogen Box.** The strain used was SFS05 (S2 Table). The growth measurements were carried out in the Eve robot (see Materials and methods) and it is given in arbitrary fluorescence units (AFU) (mean ± SD, n = 4). Cell cultures were grown in SC–met and 1.25% DMSO or 25 μM of the respective compound tested (see legend). The compounds are named according to the Pathogen box compound ID.
(DOCX)

**S11 Fig. *In vitro* growth inhibition of asexual blood stage *P. falciparum* (3D7 or Dd2) for compounds N7, N2 and PB1.** The inhibitory potential of different compounds was tested at concentrations ranging from 0.01 to 80 μM and the inhibition of parasitemia was measured after 72 hours of incubation. The growth inhibition values were expressed as percentages relative to the drug-free control and $EC_{50}$ value were calculated by plotting $log_{10}$ of compound concentrations vs growth inhibition (expressed as percentage relative to the drug-free control). The experiments were carried out in three independent assays.
(DOCX)

**S12 Fig. Cell viability analysis after treatment with compounds N2, N7 and PB.** The compounds were tested at concentrations ranging from 0.1 to 100 μM, for 24 h in MCF7 and HepG2 cells, as indicated in the panels). Cell viability (%) was calculated as the ratio between cell incubated with compound and cell incubated with DMSO. Each point represents the mean ± standard deviation, with n = 2 replicates.
(DOCX)

**S1 Table. Plasmids used in this study.**
(DOCX)

**S2 Table.** *Saccharomyces cerevisiae* **strains used in this study.**
(DOCX)

**S3 Table. Oligonucleotides used in this study.**
(DOCX)

**S4 Table. Synthetic compounds used in this study.**
(DOCX)

**S1 Text. Synthetic DHS genes with codon usage optimized for expression in** *Saccharomyces cerevisiae.*
(DOCX)

## Acknowledgments

We thank the Medicines for Malaria Venture for the gift of a Pathogen Box, and Hanna Alalam for the yeast strain HA_SC_1352control. The Swedish national infrastructure for computing (SNIC) is gratefully acknowledged for allocation of computing time on the cluster Vera at C3SE and Tetralith at NSC.

## Author Contributions

**Conceptualization:** Suélen Fernandes Silva, Angélica Hollunder Klippel, Sunniva Sigurdardóttir, Cleslei Fernando Zanelli, Per Sunnerhagen.

**Funding acquisition:** Suélen Fernandes Silva, Fabio Trindade Maranhão Costa, Katlin Brauer Massirer, Leif A. Eriksson, Cleslei Fernando Zanelli, Per Sunnerhagen.

**Investigation:** Suélen Fernandes Silva, Angélica Hollunder Klippel, Sunniva Sigurdardóttir, Catarina Bourgard, Luis Carlos Salazar-Alvarez, Heloísa Monteiro do Amaral Prado, Renan Vinicius de Araujo.

**Methodology:** Suélen Fernandes Silva, Angélica Hollunder Klippel, Sunniva Sigurdardóttir, Ievgeniia Tiukova, Heloísa Monteiro do Amaral Prado, Ross D. King, Katlin Brauer Massirer, Leif A. Eriksson, Mário Henrique Bengtson, Cleslei Fernando Zanelli, Per Sunnerhagen.

**Project administration:** Cleslei Fernando Zanelli, Per Sunnerhagen.

**Resources:** Ievgeniia Tiukova, Ross D. King, Katlin Brauer Massirer, Leif A. Eriksson, Mário Henrique Bengtson, Cleslei Fernando Zanelli, Per Sunnerhagen.

**Software:** Sayyed Jalil Mahdizadeh, Leif A. Eriksson.

**Supervision:** Sayyed Jalil Mahdizadeh, Fabio Trindade Maranhão Costa, Elizabeth Bilsland, Katlin Brauer Massirer, Leif A. Eriksson, Mário Henrique Bengtson, Cleslei Fernando Zanelli, Per Sunnerhagen.

**Visualization:** Suélen Fernandes Silva, Sunniva Sigurdardóttir, Luis Carlos Salazar-Alvarez.

**Writing – original draft:** Suélen Fernandes Silva, Sunniva Sigurdardóttir, Per Sunnerhagen.

**Writing – review & editing:** Suélen Fernandes Silva, Angélica Hollunder Klippel, Sunniva Sigurdardóttir, Sayyed Jalil Mahdizadeh, Ievgeniia Tiukova, Catarina Bourgard, Luis Carlos Salazar-Alvarez, Heloísa Monteiro do Amaral Prado, Renan Vinicius de Araujo, Fabio Trindade Maranhão Costa, Elizabeth Bilsland, Ross D. King, Katlin Brauer Massirer, Leif A. Eriksson, Mário Henrique Bengtson, Cleslei Fernando Zanelli, Per Sunnerhagen.

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
