## [Decision Letter · Decision Letter 0]

13 Oct 2024

Dear Prof. Sunnerhagen,

Thank you very much for submitting your manuscript "An experimental target-based platform in yeast for screening *Plasmodium vivax* deoxyhypusine synthase inhibitors" for consideration at PLOS Neglected Tropical Diseases. As with all papers reviewed by the journal, your manuscript was reviewed by members of the editorial board and by several independent reviewers. In light of the reviews (below this email), we would like to invite the resubmission of a significantly-revised version that takes into account the reviewers' comments. 

We cannot make any decision about publication until we have seen the revised manuscript and your response to the reviewers' comments. Your revised manuscript is also likely to be sent to reviewers for further evaluation.

Sincerely,

Shaden Kamhawi

Editor-in-Chief

Shaden Kamhawi

Editor-in-Chief

Reviewer's Responses to Questions

**Key Review Criteria Required for Acceptance?**

**Methods**

-Are the objectives of the study clearly articulated with a clear testable hypothesis stated?

-Is the study design appropriate to address the stated objectives?

-Is the population clearly described and appropriate for the hypothesis being tested?

-Is the sample size sufficient to ensure adequate power to address the hypothesis being tested?

-Were correct statistical analysis used to support conclusions?

-Are there concerns about ethical or regulatory requirements being met?

Reviewer #1: The main concern on this manuscript is the incomplete characterisation of the system.

As the pair of yeast strains expressing the human and the P. vivax DHS enzymes is presented as a tool to screen for DHS selective inhibitors, the direct quantification of the two yeast expressed DHS proteins by specific antibodies is a critical baseline parameter to a reliable measure of level and stability of the target transgenic enzymes.

Setting the system baseline at the yeast culture conditions which produce similar levels of eIF5A hypusination in the two DHS producing strain - i.e. 130 microM Methionine (human DHS) vs no Methionine (Pv DHS) (Met is needed to repress the yeast MET3 transcription factor driving transgene DHS expression) may not correspond to achieving production of similar amounts of the two DHSs. In absence of a direct quantification, the possible presence of different enzyme levels may bias the quantification of respective inhibition and the assessment of selectivity.

Also, as the system is based on measuring growth complementation of the yeast strains in which yDHS is replaced by either the human or the parasite DHS enzymes, the kinetics and extent of lethality induced by the addition of Methionine is important to evaluate reliability and comparability of the system phenotipic readout.

Reviewer #2: The author's methods are sufficient for repeating the study and the methods used are appropriate to address the hypothesis. However, Western Blotting is not robust especially with a sample size of 3. The author's used a student's T-test for statistical consideration, but did not mention the distribution of their data. The authors should run two more blots for a total n of 5 and use a Shapiro-Wilk test to determine data distribution. The authors may then accurately assess which statistical test to consider.

Reviewer #3: -Yes the objectives of the study clearly articulate the testible hypothesis mentioned.

-Yes the study desig is appropriate 

-Yes the stastical analysis is corretly applied.

-No concern about the ethical and regulatory requirments.

Reviewer #4: I would like to congratulate the authors for this brilliant research they have conducted. They have designed an experimental target-based platform in yeast for screening P. vivax deoxyhypusine synthase inhibitors. This is the only protein known to contain the amino acid hypusine essential for cell viability in eukaryotes. Inhibiting DHS is a promising strategy to develop new therapeutic alternatives.

No DHS inhibitors selective for parasite orthologs has previously been reported; They

-Genetically modified saccharomyces cerevisiae strains expressing DHS genes from Homo sapiens (HsDHS) or P. vivax DHS (Pv DHS) in place of the Wild type endogenous gene from S. cerevisiae.

They used inhibitors to assess the expression levels of these DHS genes. N2 exhibited dose-dependent inhibition of the PvDHS-expressing strain, achieving a substantial 95 % reduction in growth at 200 µM, 65 % at 100 µM, 30 % at 50 µM, and 13 % at 25 µM of the compound. In contrast, this compound did not significantly affect the growth of HsDHS (p > 0.05). Furthermore, N2 at 200 µM only led to a modest 17 % reduction in the growth of the wt strain, expressing yeast DHS. The selective growth reduction in the strain expressing PvDHS indicates that DHS is a major target of this compound.

Question: The optimization and validation of this screening platform needs to be well defined with data to validate it.

Why did the authors decide to stop at 7,5 uM for the PvDHS? The growth curve above this concentration is not shown and it would have been very important to consider other confounding factors of the parasite life stages. Will the DHS enzyme receptor have an effect on all the P. vivax life stages?

In cases of mixed infections of P. falciparium, P vivax etc will the model still be valid? All of these were to be shown in the optimization data to generate concrete and impactful conclusions.

**Results**

-Does the analysis presented match the analysis plan?

-Are the results clearly and completely presented?

-Are the figures (Tables, Images) of sufficient quality for clarity?

Reviewer #1: The baseline complementation efficiencies of the two transgenic enzymes (in 0 Met) are fairly similar (40% and 60% of the parasite vs the human DHS). In this respect, the result that a 20fold higher Met concentration is required to repress the human vs the parasite enzyme should be discussed, raising questions on the actual levels of the different DHS produced in the two strains.

Also, in the experiment testing the GC7 inhibitor on the human DHS strain, inspection of the control lanes (no GC7) shows that the intensity of the band of hypusinated eIF5A at 0 Met is virtually identical to that obtained at 130 microM Met (Figure 2B). The fact that, in contrast, the viability of the strain in the two conditions is very different (Figure 2A, no GC7 curves) questions the proposed link between yeast viability and levels of hypusilation.

Reviewer #2: All figures are of sufficient quality. Assuming appropriate statistics were used (pending the data distribution test), the authors did well with graphical representation of all figures. I am slightly concerned with figure 5, as the relative abundance does not look correct. Specifically, the wild type PB1 bands appear identical in chemiluminescence, especially when normalizing to the loading control, compared to NT and N7 treatments. Is this a fault of the representative blot? Could the authors please supply the other two blots used in a supplementary file?

Reviewer #3: -Yes the analysis match the analysis plan

-Yes tghe results are clearly and completely presented

-yes the quality of the figurs is the same.

Reviewer #4: The results are clearly presented and stem from the methods used.

However, as a question is raised in the methodology, the results are equally affected. Optimization results, and what happens in case of mixed infection.

**Conclusions**

-Are the conclusions supported by the data presented?

-Are the limitations of analysis clearly described?

-Do the authors discuss how these data can be helpful to advance our understanding of the topic under study?

-Is public health relevance addressed?

Reviewer #1: The conclusion that the described two yeast strain system is ready to screen for selective inhibitors of Plasmodium vivax DHS is overstated. The characterisation of the baseline features of the biological system requires further experiments.

Reviewer #2: The authors provide a short and sweet discussion summarizing their findings and discussing their model's future use. This is sufficient and appropriate.

Reviewer #3: Yes the conclusion support the data.

Reviewer #4: The conclusion is good and direct to the point based on the objective of the study carried out.

**Editorial and Data Presentation Modifications?**

Reviewer #1: page 2: “Shortage of in vitro culture methods” is incorrect and too generic for the Plasmodium genus, whereas it is appropriate for P. vivax.

page 3: Essentiality of DHS is overemphasized and incorrectly quoted. In reference to the pathogens (refs 14-16), an essential role is demonstrated only in the case of Trypanosome brucei; in P. falciparum, growth defects associated to the induced decreased amounts of PfDHS led to speculate of essentiality; finally, ref 16 does not contain experiments showing an essential role of DHS in the parasite studied there.

page 4: The new strain is “approx 60-fold more sensitive to the human DHS inhibitor GC7” compared to? (the parental line, presumably).

page 5: It is suggested to describe the key features of the parasite specific a.a. insertions in PvDHS compared to the yeast and the human DHSs: number, size, location in the primary sequence.

page 9: It may be useful to mention the rationale for engineering the strain with two different fluorescent reporters; the use is incidentally mentioned in reference to the robotic screening.

pages 10-11: See main comments above on the baseline features of the biological system.

page 14: The description of hypusine as “the final product of the reaction catalysed by DHS” is slightly misleading; the DHS-DOOX two step pathway to produce hypusin-eIF5a was more accurately described in the Introduction.

pag 15 – Figure 5: The remarkable increase in the level of yeast eIF5a hypusination induced by treatment with the inhibitors N2 and PB1 deserves attention and discussion.

pag 16 – Discussion: Analysis and comments on the structure of the identified hits is expected in this section. In the case of N2 and N7, in comparison with those of the described human enzyme competitive and the allosteric inhibitors. In the case of PB1, which shows some structural similarity with the allosteric inhibitor of the human DHS, a comparison with the structural binding studies proposing its anti-tubercolosis activity could be worth mentioning.

Reviewer #2: (No Response)

Reviewer #3: Minor Revision with explanation

Reviewer #4: -

**Summary and General Comments**

Reviewer #1: The manuscript describes an interesting in vivo system using two S. cerevisiae strains respectively engineered with a human and a malaria parasite (P. vivax) enzyme, DHS, which catalyses the first step of a reaction leading to a unique conserved aminoacid modification, the hypusination of EIF5a. Proposing to ultimately identify selective parasite DHS inhibitors, the manuscript presents the use of the human DHS competitive inhibitor GC7 to validate the yeast system and aims to use the failure to complement the deleted yeast DHS in the two strains as a way to screen for selective inhibitors of the parasite DHS. The conclusion that the system is ready to screen for selective inhibitors of P. vivax DHS is however overstated in the absence of the characterisation of key baseline features of the biological system.

Reviewer #2: Overall, the authors have completed what they set out to do. Figure 5 aside, they have done a great job in establishing a baseline yeast model for screening P. vivax drug targets. Given the lack of long-term in vitro culturing techniques available for vivax, this model is novel and has high impact potential.

Reviewer #3: Authors have tried to identify inhibitors of DHS from Plasmodium vivax and used genetically modified Saccharomyces cerevisiae strains expressing DHS genes from Homo sapiens (HsDHS) or P. vivax (PvDHS) in place of the endogenous DHS gene from S. cerevisiae. The manuscript is interesting but requires some explanation. Authors are suggested to respond to the comments below and resubmit the manuscript.

1. Paragraph 2 of abstract: “This new strain background was -----" which new strain is authors referring to?

2. Why authors have not proposed the same experiments on P. falciparum as the invitro culture is easy?

3. Result 3 : Establishment of S. cerevisiae……Antibody details of ScDHS missing.

4. Why were the newly developed strains not checked for expression of Hs DHS and Pv DHS in the newly developed strains are specific antibodies available for the same?

5. Discussion should be detailed out more

Reviewer #4: Authors should address the worries raised in the methodology

PLOS authors have the option to publish the peer review history of their article (what does this mean?). If published, this will include your full peer review and any attached files.

Reviewer #1: No

Reviewer #2: No

Reviewer #3: Yes: NAMRATA ANAND

Reviewer #4: Yes: Gabriel A Agbor
---

## [Editor Report · Decision Letter 1]

12 Nov 2024

Dear Prof. Sunnerhagen,

We are pleased to inform you that your manuscript 'An experimental target-based platform in yeast for screening *Plasmodium vivax* deoxyhypusine synthase inhibitors' has been provisionally accepted for publication in PLOS Neglected Tropical Diseases.

Best regards,

Shaden Kamhawi

Editor-in-Chief

Shaden Kamhawi

co-Editor-in-Chief

Paul Brindley

co-Editor-in-Chief

---

## [Editor Report · Acceptance letter]

22 Nov 2024

Dear Prof. Sunnerhagen,

We are delighted to inform you that your manuscript, "An experimental target-based platform in yeast for screening *Plasmodium vivax* deoxyhypusine synthase inhibitors," has been formally accepted for publication in PLOS Neglected Tropical Diseases.

Best regards,

Shaden Kamhawi

co-Editor-in-Chief

Paul Brindley

co-Editor-in-Chief
